# The Diagnostic Performance of Interleukin-6 and C-Reactive Protein for Early Identification of Neonatal Sepsis

**DOI:** 10.3390/diagnostics10110978

**Published:** 2020-11-20

**Authors:** Belay Tessema, Norman Lippmann, Anja Willenberg, Matthias Knüpfer, Ulrich Sack, Brigitte König

**Affiliations:** 1Institute of Medical Microbiology and Epidemiology of Infectious Diseases, Faculty of Medicine, University of Leipzig, 04103 Leipzig, Germany; Norman.Lippmann@medizin.uni-leipzig.de (N.L.); Brigitte.Koenig@medizin.uni-leipzig.de (B.K.); 2Institute of Clinical Immunology, Faculty of Medicine, University of Leipzig, 04103 Leipzig, Germany; Ulrich.Sack@medizin.uni-leipzig.de; 3Department of Medical Microbiology, College of Medicine and Health Sciences, University of Gondar, 196 Gondar, Ethiopia; 4Institute of Laboratory Medicine, Clinical Chemistry and Molecular Diagnostics, Faculty of Medicine, University of Leipzig, 04103 Leipzig, Germany; Anja.Willenberg@medizin.uni-leipzig.de; 5Department of Neonatology, Faculty of Medicine, University of Leipzig, 04103 Leipzig, Germany; Matthias.Knuepfer@medizin.uni-leipzig.de

**Keywords:** neonatal sepsis, interleukin-6, C-reactive protein, diagnosis

## Abstract

Interleukin-6 (IL-6) and C-reactive protein (CRP) are being used for diagnosis of sepsis. However, studies have reported varying cut-off levels and diagnostic performance. This study aims to investigate the optimal cut-off levels and performance of IL-6 and CRP for the diagnosis of neonatal sepsis. The study was conducted at the University Hospital of Leipzig, Germany from November 2012 to June 2020. A total of 899 neonates: 104 culture proven sepsis, 160 clinical sepsis, and 625 controls were included. Blood culture was performed using BacT/ALERT 3D system. IL-6 and CRP were analyzed by electrochemiluminescent immunoassay and immunoturbidimetric assay, respectively. Data were analyzed using SPSS 20 statistical software. Among neonates with proven sepsis, the optimal cut-off value of IL-6 was 313.5 pg/mL. The optimal cut-off values for CRP in 5 days serial measurements (CRP1, CRP2, CRP3, CRP4, and CRP5) were 2.15 mg/L, 8.01 mg/L, 6.80 mg/L, 5.25 mg/L, and 3.72 mg/L, respectively. IL-6 showed 73.1% sensitivity, 80.2% specificity, 37.6% PPV, and 94.8% NPV. The highest performance of CRP was observed in the second day with 89.4% sensitivity, 97.3% specificity, 94.5% PPV, and 98.3% NPV. The combination of IL-6 and CRP showed increase in sensitivity with decrease in specificity. In conclusion, this study defines the optimal cut-off values for IL-6 and CRP. The combination of IL-6 and CRP demonstrated increased sensitivity. The CRP 2 at cut-off 8.01 mg/L showed the highest diagnostic performance for identification of culture negative clinical sepsis cases. We recommend the combination of IL-6 (≥313.5 pg/mL) and CRP1 (≥2.15 mg/L) or IL-6 (≥313.5 pg/mL) and CRP2 (≥8.01 mg/L) for early and accurate diagnosis of neonatal sepsis. The recommendation is based on increased sensitivity, that is, to minimize the risk of any missing cases of sepsis. The CRP2 alone at cut-off 8.01 mg/L might be used to identify clinical sepsis cases among culture negative sepsis suspected neonates in hospital settings.

## 1. Introduction

Neonatal sepsis is a clinical syndrome described by systemic signs and symptoms of infection in the first month of life [1]. It is one of the major causes of mortality and morbidity during neonatal period [2,3]. Neonatal mortality due to sepsis is still 10–20% worldwide [4]. Risk factors associated to high prevalence of neonatal sepsis include the need for invasive procedures combined with immunological immaturity [5]. Early diagnosis of neonatal sepsis is still a challenge because of its non-specific clinical presentation and difficulty in differentiating from non-infectious conditions like respiratory distress syndrome or maladaptation [1,6].

Early identification and the use of broad-spectrum antibiotics remain the cornerstone of neonatal sepsis treatment. A missed identification of sepsis delays treatment and increases the risk of death [7]. On the other hand, the overuse of antimicrobial agents in patients without sepsis causes antibiotic resistance. The rapid and accurate identification of sepsis is, therefore, crucial in improving the clinical outcomes and reducing the medical costs.

The current gold standard test in diagnosis of neonatal sepsis is isolation of causative microorganisms by blood culture. However, the sensitivity and specificity of blood culture is overall low, partly because it depends on the blood volume drawn, the timing of the blood draw, any prior treatment with antibiotics, and the presence of viable organisms [8]. Hence, a negative test result for bacteremia is common and non-informative as to whether the patient needs antibiotics [9,10]. False positives due to contamination are also common and problematic [11]. Moreover, the delay in turnaround time for a blood culture result also often lead to unnecessarily long courses of empiric broad-spectrum antibiotics, exposing patients to unnecessary risk of adverse events, and leading to selection pressure for downstream antibiotic resistance.

Diagnostic biomarkers are therefore urgently needed to improve the rapid and accurate diagnosis of neonatal sepsis. An ideal diagnostic biomarker should display excellent sensitivity and negative predictive value as well as excellent specificity and positive predictive value [6]. Pro-inflammatory cytokines, acute phase proteins, adhesion molecules, cell surface markers, and chemokines are being used to identify neonatal sepsis. Interleukin-6 (IL-6) and C-reactive protein (CRP) are the two most commonly used markers for the diagnosis of neonatal sepsis.

IL-6 is one of the pro-inflammatory cytokines, and is detected in serum in the early stages of infections [12]. IL-6 has a role in the production of CRP from the liver. IL-6 level increases in early disease stages of bacterial infections, and this may be useful for the early identification of neonatal sepsis [13]. CRP is a substance that is produced in the liver and takes part in acute phase reaction. Infections and non-infectious diseases, such as malignancies and inflammatory disease, can cause increased CRP production [14]. In early stage of infection, the CRP level may be low, but serial measurements can give more helpful results and can be useful in deciding when to terminate antibiotic treatment [15]. The combination of IL-6 and CRP has recently been shown to be useful in the early diagnosis of sepsis in newborns [16]. Although studies have shown that IL-6 and CRP have diagnostic value, their cut-off values are inconstant and their diagnostic performances vary across the studies [17].

Therefore, the aims of this study are to define the optimal cut-off values for IL-6 and CRP, and to investigate the diagnostic performance of these biomarkers both in combination and alone for the early diagnosis of neonatal sepsis using large number of culture-proven sepsis, clinical sepsis, and controls.

## 2. Materials and Methods

### 2.1. Study Design and Period

We conducted this retrospective cross-sectional study at the University Hospital of Leipzig, Germany from November 2012 to June 2020. Sepsis suspected neonates with blood culture-proven sepsis, clinical sepsis, and neonates without sepsis (controls) who visited the hospital during the study period were included in this study. The demographic characteristics of neonates such as gender and age as well as the laboratory test results including blood culture results, IL-6 concentrations, and CRP concentrations were collected from the laboratory records. IL-6 and CRP concentrations were used to define the optimal cut-off levels and to evaluate the diagnostic performance of these biomarkers.

The study was conducted following the ethical considerations according to Medicinal Products law and central Ethics committee (Stellungnahme der Zentralen Ethikkommission “Die (Weiter-)Verwendung von menschlichen Körpermaterialien für Zwecke medizinischer Forschung“ (2003), and § 24 MPG II1 (2010). Only the information needed for this study was extracted and coded as is required for answering the research question. Any identifying information from the data set was removed by data controller before further usage and analysis. Safeguards were in place for appropriate and ethical use of the data. Confidentiality clauses were explicitly specified for those who do the data extraction. Permission to conduct this study was also obtained from Institute of Medical Microbiology and Epidemiology of Infectious Disease; Institute of Laboratory Medicine, Clinical Chemistry and Molecular Diagnostics; and Department of Neonatology, Faculty of Medicine, University of Leipzig.

### 2.2. Classification of Study Participants

Sepsis-suspected neonates were retrospectively classified as proven sepsis, clinical sepsis, or controls based on C-reactive protein (CRP) and blood culture results. Proven sepsis was defined as CRP > 10 mg/L in at least one of the five serial measurements and positive blood culture. Clinical sepsis was defined as CRP > 10 mg/L in at least one of the five serial measurements and negative blood culture. No sepsis (control) was defined as neonates suspected for sepsis, with negative blood culture, CRP < 10 mg/L in all five serial measurements, and neonates who had not started antibiotics treatment before blood collection. Neonates with positive blood cultures for coagulase negative staphylococci (CoNS) organisms and CRP < 10 mg/L in all five serial measurements were considered as potential contamination and excluded from our analysis. Late onset sepsis was defined as sepsis onset > 72 h of neonates after birth. Early onset sepsis was defined as sepsis onset ≤ 72 h of neonates after birth [18,19].

In case of suspicion of sepsis, the neonatologist on duty took venous blood sample on day 1 for CRP, IL-6, and blood culture. Additional venous or capillary blood samples depending on the volume of blood required for the test were collected on day 2, 3, 4, and 5 for CRP measurements.

### 2.3. Blood Culture

From sepsis-suspected neonate, 0.5–1 mL of blood was collected, inoculated into BacT/ALERT bottles and loaded into the automated BacT/ALERT 3D system. Bottles were incubated up to 7 days at 37 °C. Once the positive signal was detected in the blood culture systems, 1 mL was drawn out for Gram staining and subculture. Subculture was routinely performed using a blood agar plate, Endo agar, chocolate agar, Bile Esculin agar, Sabouraud dextrose agar for aerobic culture and Brucella agar for anaerobic culture. Colonies of bacteria or yeasts were identified with Vitek matrix-assisted laser desorption ionization time-of-flight (MALDI-TOF) mass spectrometry (BioMérieux, Marcy L’Etoile, France).

### 2.4. IL-6 and CRP Determinations

Blood samples were transported from neonatology department to laboratory using automated tube mail system immediately after collection. Blood samples were centrifuged to separate the serum and processed for the test as soon as they arrived at the laboratory. Serum IL-6 concentration was measured by electrochemiluminescent immunoassay (ECLIA) following the manufacturer’s instructions (Cobas; Roche Diagnostics, Mannheim, Germany). The detection limit of this method for IL-6 is 2.5 pg/mL. Serial measurements of serum CRP concentrations were performed in the first five days after sepsis was suspected by Immunoturbidimetric assay, Tina-quant C-Reactive Protein method according to the manufacturer’s instructions (Cobas; Roche Diagnostics, Mannheim, Germany). The detection limit of this method for CRP is 0.3 mg/L.

### 2.5. Statistical Analyses

Data were entered and analyzed using SPSS statistical package version 20 software. Data were checked for normal distribution using Skewness and Kurtosis Z-values, the Shapiro–Wilk test *p*-value, and visual outputs including histograms, normal Q-Q plots, and Box plots. For data that are not normally distributed, non-parametrical tests were used and median and range were presented. Descriptive statistics including frequency and percentages of culture proven sepsis cases, clinical sepsis, and no sepsis controls, and organisms isolated from culture positive neonates were calculated. The mean and standard deviation (SD) were calculated to assess the mean age of neonates. The median and range were calculated for the concentrations of IL-6 and CRP. Receiver-operating characteristics (ROC) curve was drawn to calculate the area under the curve (AUC) and coordinates of the curve. The optimal cut-off levels of IL-6 and CRP were determined based on the results of sensitivity and 1-specificity of the coordinates of the ROC curve. Cross tabulation was used to calculate the sensitivity, specificity, positive predictive value (PPV), and negative predictive value (NPV) for IL-6 and CRP tests in combination or alone.

## 3. Results

A total of 916 neonates with culture, IL-6, and CRP results were recorded during the study period. However, 17 culture positive neonates with CoNS organisms were excluded from the final analysis due to suspicion of contamination as the concentration of CRP was <10 mg/L in all five serial measurements. Finally, a total of 899 neonates were included in this study. Of which, 104 (11.6%) neonates were blood culture proven sepsis cases, 160 (17.8%) were clinical sepsis cases, and 635 (70.6%) were non sepsis controls. Majority, 525 (58.4%) of the neonates were male. The mean ± SD age of neonates was 3.9 ± 6.5 days, and 645 (70.1%) neonates were in the age group of ≤72 h. Among neonates with culture proven sepsis, 95 (91.3%) were late onset sepsis cases and the mean ± SD age of the neonates was 11.4 ± 7.3 days. Among clinical sepsis cases, 87 (54.4%) were early onset sepsis cases and the mean ± SD age was 7.2 ± 8.6 days (Table 1).

The pathogens identified in the positive blood cultures and the corresponding median and range of IL-6 and CRP concentrations are presented in Table 2. A total of 16 types of microorganisms were isolated from blood cultures. CoNS species was more frequently isolated, 53 (51%) than the non CoNS microorganisms. The most predominant sepsis-causing organism was *Staphylococcus epidermidis,* 38 (36.5%), followed by *Escherichia coli,* 20 (19.2%)*, Staphylococcus haemolyticus,* 12 (11.5%), and *Staphylococcus aureus*, 11 (10.6%).

Non CoNS isolates showed markedly higher median and range concentrations of IL-6, and CRP in five days serial measurements compared with CoNS. In culture proven sepsis cases, the highest concentration of CRP was observed in the second day measurement (CRP2). The CRP concentration showed a declining trend after the second day of measurement (CRP 2) in the third (CRP3), fourth (CRP4), and fifth days (CRP5). The median concentration of IL-6 among proven sepsis (1612 pg/mL) was significantly higher than that of the clinical sepsis (138 pg/mL) and control groups (57 pg/mL) (*p* < 0.001). Similarly, the median concentration of IL-6 among clinical sepsis group was significantly higher than that of the control group (*p* < 0.001). The median CRP1 concentration among proven (12 mg/L) and clinical sepsis (21 mg/L) groups was significantly higher than that of the control group (0.6 mg/L) (*p* < 0.001). However, the median CRP1 concentration among proven sepsis group were not significantly different from that of the clinical sepsis group (data not shown).

*Pseudomonas aeruginosa, Enterobacter hormaechei,* and *Enterobacter cloacae. Listeria monocytogenes-* and *E.coli*-infected neonates showed higher median concentrations of IL-6 compared with neonates infected by other organisms. Similarly, neonates infected by *Listeria monocytogenes*, *Morganella morganii,* and *Candida albicans* had higher median concentrations of CRP (in all the five days serial measurements) compared with neonates infected by other isolates. Among CoNS, *Staphylococcus haemolyticus* showed the highest median concentrations of Il-6 and CRP (CRP1 and CRP2) compared with neonates infected by other CoNS isolates.

ROC curves of blood culture proven sepsis cases versus controls are shown in Figure 1. The AUC value for IL-6 was 0.88 (95% CI: 0.849–0.910). This indicates the accuracy of IL-6 to correctly discriminate sepsis cases from controls was good. The AUC values for the five days serial measurements of CRP showed excellent accuracy: CRP1, 0.905 (0.870–0.940), CRP2, 0.998 (0.995–1), CRP3, 0.995 (0.991–0.999), CRP4, 0.986 (0.976–0.996), and CRP5, 0.995 (0.991–0.999).

The optimal cut-off values, diagnostic sensitivity, specificity, PPV, and NPV of IL-6 and CRP of proven sepsis versus controls are presented in Table 3. The optimal cut-off levels of IL-6 and the five days serial measurements of CRP were identified based on the results of sensitivity and 1-Specificity of the coordinates of the ROC curve. The optimal cut-off value of IL-6 among proven sepsis versus controls was 313.5 pg/mL. The optimal cut-off values of the 5 days serial measurements of CRP among proven sepsis: CRP1, CRP 2, CRP3, CRP4, and CRP5 were 2.15 mg/L, 8.01 mg/L, 6.80 mg/L, 5.25 mg/L, and 3.72 mg/L, respectively.

The sensitivity, specificity, PPV, and NPV of IL-6 (313.5 pg/mL) were 73.1%, 80.2%, 37.6%, and 94.8%, respectively. The highest performance of CRP for the diagnosis of neonatal sepsis was observed at the second day of serial measurements, CRP2 (8.01 mg/L) with sensitivity (89.4%), specificity (97.3%), PPV (84.5%), and NPV (99.3%). CRP 1 at low optimal cut-off value (2.15 mg/L) also showed good diagnostic performance with sensitivity (83.7%), specificity (82.2%), PPV (43.5%), and NPV (96.8%). The combination of IL-6 with CRP 1, CRP 2, CRP3, CRP 4, or CRP 5 showed superior sensitivity with a slight decrease in specificity than the use of IL-6 or CRP alone for the diagnosis of neonatal sepsis.

The diagnostic performance of IL-6, CRP 1, and CRP 2 at their optimal cut-off values for identification of sepsis among culture negative clinical sepsis cases is shown in Table 4. The highest diagnostic performance was observed for CRP 2 at cut-off 8.01 mg/L with sensitivity (90.4%), specificity and PPV (100%), and NPV (97.3%). The lowest sensitivity (32.3%) and PPV (37.5%) were observed for IL-6 at cut-off 313,5 pg/mL. However, its specificity and NPV were 83.6% and 80.2%, respectively.

## 4. Discussion

In this study, we attempt to define the optimal cut-off values of serum IL-6 and CRP using ROC curves, and evaluate their performance for early diagnosis of neonatal sepsis. The clinical signs and symptoms of neonatal sepsis are usually non-specific, and difficult to differentiate from non-infectious conditions. Although blood culture remains the gold standard, its long turnaround time, false-negative results, and low culture positivity rates remain a major challenge in the diagnosis of neonatal sepsis [20]. The ideal diagnostic marker is desired to have about 100% sensitivity (infected neonates have a positive test) and NPV (a negative test confidently rules out infection). To minimize the unnecessary use of antibiotics in false-positive cases, a diagnostic marker also needs to have reasonably high, preferably better than 85% specificity (the test is negative if infection is absent), and a good PPV (infection is present when the test is positive) [4].

Several biochemical and immunological markers increase in the plasma during neonatal sepsis, such as increased CRP, IL-6, TNF-α, procalcitonin, and E-selectin [21,22,23,24]. IL-6 and CRP are being used most commonly for the diagnosis of neonatal sepsis. IL-6 is a vital cytokine of the early host response to infection. Its concentration rises abruptly after exposure to bacterial products and precedes the rise in CRP. It has a very short half-life, and the concentration falls quickly with treatment, becoming undetectable in most infected patients within 24 h. CRP is synthesized by the liver within 6–8 h in an inflammatory response, peaks at 24–48 h, and then reduces over time as the inflammation resolves [25]. In the previous studies, the optimal cut-off values of IL-6 and CRP are defined inconsistently, and their diagnostic performance for identification of neonatal sepsis varies.

Our study included large number of neonates with culture proven sepsis, clinical sepsis, and controls to correctly define the optimal cut-off values and measure the diagnostic performances of IL-6 and CRP. To our knowledge, this is the largest study conducted using 899 neonates. Sample sizes in the previous studies ranged from 9 to 332 neonates (17, 23). The large sample size and use of automated blood culture system are the foremost strengths of our study.

The results of the present study showed a significant increase (<0.001) in IL-6 and CRP concentrations among culture proven sepsis and clinical sepsis cases compared to controls. We also observed remarkable differences in the cut-off values and diagnostic performance of CRP in the first five days of serial measurements after neonates are suspected for sepsis. The optimal cut-off level for IL-6 in neonates with culture proven sepsis was 313.5 pg/mL. At this cut-off level, we observed 73.1% sensitivity, 80.2% specificity, 37.6% PPV, and 94.8% NPV. Other previous studies have calculated varying cut-off levels and reported inconsistent sensitivity, specificity, PPV, and NPV for IL-6. The cut-off values for IL-6 have been shown to be from 10 to 500 pg/mL, with most falling from 10 to 30 pg/mL [26]. A previous study conducted among 34 neonates with culture confirmed and clinical sepsis reported a cut-off value for IL-6 at 20 pg/mL, with 91% sensitivity, 74% specificity, 78% PPV, and 89% NPV [27]. A study conducted by Ng et al. among 45 neonates with culture confirmed sepsis reported a cut-off level for IL-6 at 31 pg/mL, with 89% sensitivity, 96% specificity, 95% PPV, and 91% NPV [28]. In another study, a cut-off level for IL-6 was 21.5 pg/mL, with 75% sensitivity, 82% specificity, 92% PPV, and 52% NPV for neonates with culture confirmed sepsis [25].

In our study, five different optimal cut-off levels were observed for CRP measured in the first five days of serial measurements among proven sepsis. The highest performance of CRP for the diagnosis of neonatal sepsis was observed in the second day measurement with 89.4% sensitivity, 97.3% specificity, 84.5% PPV, and 98.3% NPV. CRP concentration in the first day of sepsis symptoms is believed to be insufficient for the diagnosis of sepsis; however, this study demonstrated that CRP 1 at low optimal cut-off value (2.15 mg/L) has good diagnostic performance with sensitivity (83.7%), specificity (82.2%), PPV (43.5%), and NPV (96.8%). Other previous studies reported variable CRP cut-off levels between 6 and 10 mg/L for a single day measurement without serial measurements. A study done using cord blood reported a cut-off level for CRP at 6 mg/L, with 80% sensitivity, 60% specificity, 7.7% PPV, and 98.6% NPV [29]. A study done by Benitz et al. gave a 10 mg/L CRP cut-off, with 78% sensitivity, 78% specificity, 67% PPV, and 97% NPV [30]. A 4 mg/L CRP cut-off level was also reported with 95.7% sensitivity, 88.9% specificity, 78.6% PPV, and 98% NPV [31]. In another study, a cut-off level for CRP was 5.82 mg/L with 71% sensitivity, 97% specificity, 99% PPV, and 49% NPV [25].

A previous study recommended the combination of IL-6 (>70 pg/mL) and CRP (>10 mg/L) to achieve 91% sensitivity, 74% specificity, 43% PPV, and 98% NPV [17]. Another study proposed the combination of IL-6 (>24.65 pg/mL) and CRP (>5.82 mg/L) to attain adequate diagnostic performance [25]. We recommend the combination of IL-6 (≥313.5 pg/mL) and CRP 1 (≥2.15 mg/L) or CRP 2 (≥8.01 mg/L) for early and accurate diagnosis of neonatal sepsis. The diagnostic performance of these combinations is higher than the use of IL-6 or CRP alone as these combinations showed markedly increased sensitivity and NPV with a slightly decreased specificity and PPV compared with the use of IL-6 or CRP alone for the diagnosis of neonatal sepsis.

Variations in the cut-off levels and the diagnostic performance of IL-6 and CRP among the previous studies and this study might be due to the variations in the number of neonates enrolled in the study, the type of patient categories (culture proven sepsis, clinical sepsis or both) used for cut-off level calculations, the age of the neonates and the number of low birth weight infants included in the study. Sampling time is also another important issue in the correct diagnosis of neonatal sepsis. This issue was clearly demonstrated in our study with marked differences in CRP concentrations, differences in optimal cut-off values and diagnostic performance in the five days serial measurements. The peak time of CRP concentration was observed in the second day of inflammatory response and afterwards the concentration showed a declining trend in the third, fourth, and fifth days of measurements. In most other previous studies optimal cut-off values calculated only based on a single measurement of CRP. There might also be differences for IL-6 and CRP results between different laboratories, especially if they use different assays.

In our study, the most frequently isolated organisms among culture positive cases were CoNS species and the most predominant sepsis-causing organism was *Staphylococcus epidermidis* (38.8%). Similarly, other previous studies showed that CONS organisms were the most common organisms among culture positive neonates [32,33]. A previous study suggested the association of CoNS with the biofilm forming strains that inhibits the host immune system and enhances the risk of counteracting the infection [33]. External factors, such as the disruption of skin barrier by medical devices and the selective pressure due to antibiotics, could contribute to the conversion of *S. epidermidis* from a member of the skin microflora to an infectious pathogen. It is also commonly believed that the vulnerability of neonates to nosocomial sepsis caused by *S. epidermidis* infections arises, at least in part, from an immature immunity marked by deficiencies in numerous components of the immune system [34].

In this study, neonates infected by non-CoNS showed higher median concentrations of IL-6 and CRP compared with CoNS. Neonates infected by *Pseudomonas aeruginosa, Enterobacter hormaechei*, *Enterobacter cloacae, Listeria monocytogenes,* and *E.coli* showed the highest median concertations of IL-6 compared with other organisms. The highest median concentrations of CRP was also observed among neonates infected by *Listeria monocytogenes*, *Morganella morganii,* and *Candida albicans. Staphylococcus haemolyticus* showed the highest median concentrations of Il-6 and CRP compared with neonates infected by other CoNS isolates. These variations in the median concentrations of IL-6 and CRP among neonates infected by different microorganisms might be due to the difference in the host inflammatory response during infection by different microorganisms.

The CRP 2 at cut-off 8.01 mg/L showed the highest diagnostic performance for identification of culture negative clinical sepsis cases with sensitivity (90.4%), specificity and PPV (100%), and NPV (97.3%). Therefore, CRP2 at this cut-off value might be used to identify clinical sepsis cases among culture negative sepsis suspected neonates in hospital settings.

### Limitations

One of the methodological limitations in this study is the lack of clinical data such as signs and symptoms of neonates, gestational age, and mode of delivery. Moreover, other useful data to more accurately define clinical sepsis and control groups such as the data from automethod blood culture, enzymology, PCR, and ethnicity are not available in the laboratory records. These limitations are due to the retrospective design of our study, however, the findings are strengthen by relatively large number of investigated neonates, the use of automated blood culture system for all neonates, and the use of culture proven neonates to define the optimal cut-off values for IL-6 and CRP.

## 5. Conclusions

This study defines the optimal cut-off values for IL-6 and CRP in the first five days of serial measurements using large number of culture-proven sepsis cases. Of the five days serial measurements of CRP, the CRP measured in the second day showed the highest median concentration and the highest diagnostic performance. CRP1 at low optimal cut-off value showed good diagnostic performance. The combination of IL-6 and CRP demonstrated increased sensitivity. The CRP 2 at cut-off 8.01 mg/L showed the highest diagnostic performance for identification of culture negative clinical sepsis cases. We recommend the combination of IL-6 (≥313.5 pg/mL) and CRP1 (≥2.15 mg/L) or IL-6 (≥313.5 pg/mL) and CRP2 (≥8.01 mg/L) for early and accurate diagnosis of neonatal sepsis. The recommendation is based on increased sensitivity, that is, to minimize the risk of missing any cases of sepsis. The CRP2 alone at cut-off 8.01 mg/L might be used to identify clinical sepsis cases among culture negative sepsis-suspected neonates in hospital settings.

## Figures and Tables

**Figure 1 diagnostics-10-00978-f001:**
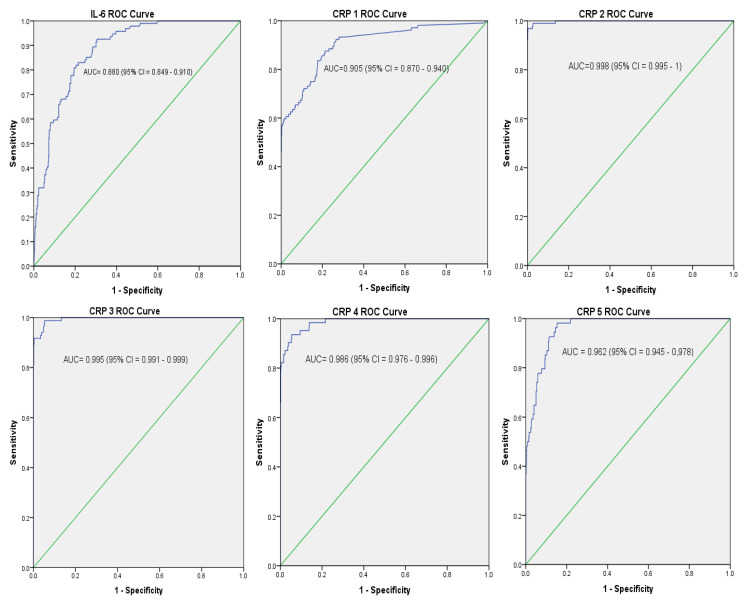
The receiver operating characteristic (ROC) curves for IL-6 and CRP tests drawn from sensitivity versus 1-specificity using culture proven sepsis cases and controls. ROC shows the performance of a test at all classification thresholds. AUC = area under the ROC curve. AUC provides an aggregate measure of performance of a test across all possible classification thresholds. AUC indicates the accuracy of IL-6 or CRP to correctly classify sepsis cases from controls. An AUC of 1 represents a perfect test; 0.90–1 = excellent, 0.80–0.90 = good; 0.70–0.80 = fair; 0.60–0.70 = poor; 0.50–0.60 = fail and an AUC of 0.5 represents a worthless test. CI = Confidence interval.

**Table 1 diagnostics-10-00978-t001:** Demographic characteristics of neonates according to the categories.

Characteristics	Categories			Total *n* (%)
	Proven Sepsis *n* (%)	Clinical Sepsis *n* (%)	Controls *n* (%)	
**Total**	104 (11.6)	160 (17.8)	635 (70.6)	899 (100)
**Gender**				
Male	56 (53.8)	93 (58.1)	376 (59.2)	525 (58.4)
Female	48 (46.2)	67 (41.9)	259 (40.8)	374 (41.6)
**Age groups**				
≤72 h	9 (8.7)	87 (54.4)	549 (86.5)	645 (71.7)
>72 h	95 (91.3)	73 (45.6)	86 (13.5)	254 (28.3)
**Age in days** (**mean** ± **SD**)	11.4 ± 7.3	7.2 ± 8.6	2.5 ± 5.0	4.4 ± 6.8

*n* = number; SD = standard deviation.

**Table 2 diagnostics-10-00978-t002:** Frequency of microorganisms isolated from blood culture positive neonates, and the median and range of IL-6 and CRP concentrations.

Microorganism	Frequency *n* (%)	IL-6 pg/mL Median(range)	CRP mg/L Median(range)				
			CRP1	CRP2	CRP3	CRP4	CRP5
CoNS	53 (51.0)	777 (67–50000)	12 (0.6–100)	40 (3.3–131)	23 (6.5–154)	15 (1.7–138)	8 (1.6–132)
*Staphylococcus* *epidermidis*	38 (36.5)	663 (97–42285)	11 (1–90)	33 (8.1–131)	19 (6.5–91)	12 (1.7–66)	7 (1.6–32)
*Staphylococcus* *haemolyticus*	12 (11.5)	2692 (67–50000)	21 (0.6–100)	56 (17.7–112)	33 (13.7–154)	16 (12.9–138)	10 (4.4–132)
*Staphylococcus* *hominis*	3 (2.9)	1308 (162–1695)	4 (0.6–31)	43 (3.3–83)	48 (13.6–82)	35 (19.2–50)	31 (4.0–45)
Non CoNS	51 (49.0)	3310 (37–50000)	14 (0.3–220)	46 (8.0–200)	43 (3.6–229)	28 (3.3–103)	14 (3.1–105)
*Escherichia coli*	20 (19.2)	25264 (54–50000)	12 (0.4–90)	45 (8.0–200)	46 (18.1–144)	21 (3.3–89)	20 (6.4–82)
*Staphylococcus aureus*	11 (10.6)	600 (193–13074)	24 (1.1–174)	43 (13.8–194)	20 (7.4–229)	25 (13.1–103)	7 (3.1–64)
*Enterococcus faecalis*	5 (4.8)	1149 (180–50000)	6 (0.3–24)	27 (16.0–54)	37 (3.6–67)	19 (11.4–76)	5 (3.9–82)
*Streptococcus agalactiae*	3 (2.9)	374 (37–3310)	6 (5.1–120)	51 (18.7–61)	25 (10.4–41)	28	5
*Enterobacter cloacae*	2 (1.9)	35205 (20410–50000)	3 (0.8–4)	62 (16.0–108)	98 (79.0–117)	40 (32.4–48)	22 (4.0–40)
*Listeria monocytogenes*	2 (1.9)	33915 (17830–50000)	210 (199.6–220)	171 (165.2–176)	86 (80.6–90)	38 (34.8–40)	*n*/A
*Klebsiella oxytoca*	2 (1.7)	5188 (1049–9326)	7 (1.9–12)	43 (27.8–57)	*n*/A	8	*n*/A
*Bacillus cereus*	1 (1.0)	1695	31	83	82	50	31
*Klebsiella pneumoniae*	1 (1.0)	2216	22	38	18	11	7
*Enterobacter hormaechei*	1 (1.0)	50000 *	88	160	105	65	50
*Morganella morganii*	1 (1.9)	*n*/A	169	194	229	85	64
*Pseudomonas aeruginosa*	1 (1.9)	50000 *	2	n/A	n/A	n/A	n/A
*Candida albicans*	1 (1.0)	368.0	103	137	136	96	105
Total	104 (100)	1612 (37–50000)	12 (0.3–220)	44 (3.3–200)	28 (3.6–229)	19 (1.7–138)	9 (1.6–132)

*n* = number; CoNS = Coagulase negative staphylococci; IL-6 = Interleukin 6; CRP= C-reactive protein; CRP1, CRP2, CRP3, CRP4, CRP5= CRP measured in the first, second, third, fourth and fifth days, respectively, after the neonate was suspected for sepsis; n/A = not available; * = samples with concentration > 50000 pg/mL are usually reported as >50000 pg/mL. For these two neonates, the IL-6 concentration was recorded as >50000 pg/mL. For statistical analysis, these results were coded as 50000 pg/mL. Therefore, the median IL-6 concentration 50000 pg/mL indicates concentration > 50000 pg/mL.

**Table 3 diagnostics-10-00978-t003:** Optimum cut-off values and diagnostic sensitivity (SN), specificity (SP), positive predictive value (PPV), and negative predictive value (NPV) of IL-6 and CRP.

Patient Category	Single and Combine Test ^#^	Cut-off Value *	AUC (95% CI)	SN (%)	SP (%)	PPV (%)	NPV (%)
Proven sepsis versus Controls	IL-6	313.5 pg/mL	0.88 (0.85–0.91)	73.1	80.2	37.6	94.8
	CRP 1	2.15 mg/L	0.91 (0.87–0.94)	83.7	82.2	43.5	96.8
	IL-6 and CRP 1	-	-	98.1	69.0	34.1	99.5
	CRP 2	8.01 mg/L	1.00	89.4	97.3	84.5	98.3
	IL-6 and CRP 2	=	-	97.1	78.1	42.1	99.4
	CRP 3	6.80 mg/L	1.00 (0.99–1.00)	76.9	95.3	72.7	96.2
	IL-6 and CRP 3	-	-	91.3	76.7	39.1	98.2
	CRP 4	5.25 mg/L	0.99 (0.98–1.00)	55.8	92.3	54.2	92.7
	IL-6 and CRP 4	-	-	88.5	74.8	36.5	97.5
	CRP 5	3.72 mg/L	0.96 (0.95–0.98)	48.1	88.2	40.0	60.0
	IL-6 and CRP 5	-	-	85.6	73.1	34.2	96.9

^#^ = the combination of IL-6 and CRP test results were interpreted as positive when at least one of the two tests (IL-6 or CRP) was ≥ the cut-off value. CRP1, CRP2, CRP3, CRP4, CRP5 = CRP measured in the first, second, third, fourth and fifth days, respectively, after the neonate was suspected for sepsis. * = Cut-off values were determined based on the sensitivity and 1-spesificity of the coordinates of the ROC curve. AUC = area under the ROC curve. CI = confidence interval.

**Table 4 diagnostics-10-00978-t004:** The diagnostic performance of IL-6 and CRP at their optimal cut-off values for identification of sepsis among culture negative clinical sepsis cases.

Patient Category	Single and Combine Test ^#^	SN (%)	SP (%)	PPV (%)	NPV (%)
Clinical sepsis vs. Controls	IL-6 (313.5 pg/mL)	32.3	83.6	37.5	80.2
	CRP 1 (2.15 mg/L)	58.6	100	100	82.2
	CRP 2 (8.01 mg/L)	90.4	100	100	97.3
	IL-6 and CRP 1	44.8	100	100	69
	IL-6 and CRP 2	53.5	100	100	78.1

^#^ = the combination of IL-6 and CRP test results were interpreted as positive when at least one of the two tests (IL-6 or CRP) was ≥ the cut-off value. CRP1 and CRP2 = CRP measured in the first and second days, respectively, after the neonate was suspected for sepsis. SN = sensitivity, SP = specificity, PPV = positive predictive value, and NPV = negative predictive value.

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
