# Peer review of "The Diagnostic Performance of Interleukin-6 and C-Reactive Protein for Early Identification of Neonatal Sepsis"

_diagnostics, 2020, doi:10.3390/diagnostics10110978_

Round 1

Reviewer 1 Report

Peer review

Title: The diagnostic performance of interleukin-6 and C-reactive protein for early identification of neonatal sepsis

This is a retrospective study of the diagnostic value of using CRP and IL-6 to correctly identify neonates with sepsis. Blood culture was used as gold standard. The aim is to find optimal cut-offs for CRP and IL-6 to diagnose neonatal sepsis. These markers for sepsis are far from new, but as the authors also claim, there is a need of defining sensitive and specific cut-offs for neonates, especially since it is clinically difficult to identify sepsis and that adequate and early treatment is extremely important.  The aim of the study is thus highly important for the neonatal care. The study also includes a rather large number of neonates (n=899) of which 104 had culture proven sepsis. A strength of the study is also that CRP was measured serially day 1 to 5 after the suspicion of sepsis. Limitations of the study are the lack of information on gestational date, prematurity and clinical data. The age in days also differ considerably between neonates with sepsis compared to controls.

Pleas respond to the following questions and remarks.

Abstract:

Introduction:

Line 58: Please correct the sentence to: Moreover, the delay in turnaround time for a blood culture result also…

Line 71: This sentence in unclear since you compare two different parameters “IL-6 levels may be greater than CRP…” Please, reformulate so that CRP and IL-6 are related to something, such as reference range or relative increase.

Materials and Methods:

2.1 Please give a more detailed description of the control group. What were the inclusion criteria? Where all of these neonates suspected for sepsis? Where these infants hospitalized and what diseases were they admitted for?

Please specify when the sample for IL-6 and blood culture analysis was taken (Day 1 or others?)

2.4 Please give more details about the sampling procedure. Venous or capillary samples? How long was the delay between sampling and centrifugation at the laboratory? This information is important since IL-6 will increase in whole blood when stored in RT (see Gong Y et al. J Clin Lab Anal, 2019).

Please specify which CRP method was used. Was it a high sensitivity method? What was the detection limit?

Please also specify the detection limit of the IL-6 method.

2.5

The data should be checked for normal distribution. Specify which statistical method was used. If data is not normally distributed please use non-parametrical tests and present median (not mean) and interquartile range or range (not SD).

Line 90: Change to: “who visited the hospital…”

Line 102: Explain the abbreviation CoNS.

Results and tables:

Table 2: Since the number of cases infected with certain strains of microorganisms are very low for most groups IL-6 and CRP results should be presented as median and range or only median (not mean and SD).

Line 162: Please add information on CRP and IL-6 in the control group and the group with clinical sepsis to be compared with the group with culture proven sepsis, presented as mean/median and SD/range (depending on distribution fitting).

Table 3: It is somewhat problematic to include the group with clinical sepsis in the validation of optimal cut-offs, since patients in this group by definition are diagnosed with sepsis because of increased CRP (circular proof). I suggest instead to test SN, SP, PPV and NPV for this group separately using cut-offs for proven sepsis.

Discussion:

The suggested cut-off value for IL-6 is relatively high compared to previous studies. Is there an explanation of this? Please comment on this.

Line 340: Specify what is meant by “The combination of IL-6 and CRP has better diagnostic performance than the use of IL-6 or CRP alone”. This combination will give a better sensitivity, but at a cost of a decreased specificity and PPV.

Additionally, changes may be needed after recalculating data.

Ethics:

There is no declaration that the study was performed according to the rules of the Declaration of Helsinki and no information on approval from an ethics committee. This must be added according to Instructions of Authors.

Author Response

Reviewer 1

This is a retrospective study of the diagnostic value of using CRP and IL-6 to correctly identify neonates with sepsis. Blood culture was used as gold standard. The aim is to find optimal cut-offs for CRP and IL-6 to diagnose neonatal sepsis. These markers for sepsis are far from new, but as the authors also claim, there is a need of defining sensitive and specific cut-offs for neonates, especially since it is clinically difficult to identify sepsis and that adequate and early treatment is extremely important.  The aim of the study is thus highly important for the neonatal care. The study also includes a rather large number of neonates (n=899) of which 104 had culture proven sepsis. A strength of the study is also that CRP was measured serially day 1 to 5 after the suspicion of sepsis. Limitations of the study are the lack of information on gestational date, prematurity and clinical data. The age in days also differ considerably between neonates with sepsis compared to controls.

Please respond to the following questions and remarks.

Response: The reviewer’s comments are greatly appreciated. Authors responded to the questions and remarks raised by the reviewer. The manuscript has been revised according to the reviewer’s comments and suggestions.

Abstract:

Introduction:

Comment 1: Line 58: Please correct the sentence to: Moreover, the delay in turnaround time for a blood culture result also…

Response:  Corrected: Line 59.

Comment 2: Line 71: This sentence in unclear since you compare two different parameters “IL-6 levels may be greater than CRP…” Please, reformulate so that CRP and IL-6 are related to something, such as reference range or relative increase.

Response: The sentence is reformulated: Line 71 and 72.

Materials and Methods:

Comment 1: 2.1: Please give a more detailed description of the control group. What were the inclusion criteria?

Response:  Control group was defined as neonates suspected for sepsis, with negative blood culture, CRP < 10 mg/l in all five serial measurements and neonates who had not started antibiotics treatment before blood collection. Line 114-116.

Where all of these neonates suspected for sepsis?

Response: Yes. Line 110.

Where these infants hospitalized and what diseases were they admitted for?

Response: There is no information in the patients’ laboratory records whether the neonates were hospitalized or not.

Please specify when the sample for IL-6 and blood culture analysis was taken (Day 1 or others?)

Response: In case of suspicion of sepsis, the neonatologist on duty took venous blood sample on day 1 for CRP, IL-6 and blood culture. Line 122 and 124.

Comment 2: 2.4: Please give more details about the sampling procedure. Venous or capillary samples? How long was the delay between sampling and centrifugation at the laboratory? This information is important since IL-6 will increase in whole blood when stored in RT (see Gong Y et al. J Clin Lab Anal, 2019).

Response: In case of suspicion of sepsis, the neonatologist on duty took venous blood sample on day 1 for CRP, IL-6 and blood culture. Additional venous or capillary blood samples depending on the volume of blood required for the test were collected on day 2, 3, 4 and 5 for CRP measurements. Blood samples were transported from neonatology department to laboratory using automated tube mail system immediately after collection. Blood samples were centrifuged to separate serum and processed for the test as soon as they arrived at the laboratory. Line 122-124, and line 137-139.

Please specify which CRP method was used.

Response: Tina-quant C-Reactive Protein method. Line 143 and 144.

Was it a high sensitivity method? What was the detection limit?

Response: The detection limit of this method for CRP is 0.6- 700 mg/l. line 145.

Please also specify the detection limit of the IL-6 method.

Response: The detection limit of this method for IL-6 is 2.5 to 50000 pg/ml. line 141 and 142.

Comment 3: 2.5: The data should be checked for normal distribution. Specify which statistical method was used. If data is not normally distributed please use non-parametrical tests and present median (not mean) and interquartile range or range (not SD).

Response: Data were checked for normal distribution using Skewness and Kurtosis Z-values, the Shapiro-Wilk test P-value, and visual outputs including histograms, normal Q-Q plots and Box plots. For data which are not normally distributed, non-parametrical tests were used and median and range were presented. Line 148-152.

Line 90: Change to: “who visited the hospital…”

Response:  Corrected: line 91 and 92.

Line 102: Explain the abbreviation CoNS.

Response: Coagulase negative staphylococci (CoNS). Line 117.

Results and tables:

Comment 1: Table 2: Since the number of cases infected with certain strains of microorganisms are very low for most groups IL-6 and CRP results should be presented as median and range or only median (not mean and SD).

Response: Results presented as median and range or only median. Line 206-217.

Comment 2: Line 162: Please add information on CRP and IL-6 in the control group and the group with clinical sepsis to be compared with the group with culture proven sepsis, presented as mean/median and SD/range (depending on distribution fitting).

Response: The median concentration of IL-6 among proven sepsis was significantly higher than that of the clinical sepsis and control groups (P<0.001). Similarly, the median concentration of IL-6 among clinical sepsis group was significantly higher than that of the control group (P<0.001). The median CRP concentrations in all serial measurements among proven and clinical sepsis groups were significantly higher than that of the control group (P<0.001). However, the median CRP concentrations among proven sepsis group were not significantly different from that of the clinical sepsis group (data not shown).Line 187-195.

Comment 3: Table 3: It is somewhat problematic to include the group with clinical sepsis in the validation of optimal cut-offs, since patients in this group by definition are diagnosed with sepsis because of increased CRP (circular proof). I suggest instead to test SN, SP, PPV and NPV for this group separately using cut-offs for proven sepsis.

Response: The group with clinical sepsis in the validation of optimal cut-offs is excluded from Table 3. The test for SN, SP, PPV and NPV for this group is presented separately in Table 4. Line 256-284.

Discussion

Comment 1: The suggested cut-off value for IL-6 is relatively high compared to previous studies. Is there an explanation of this? Please comment on this.

Response: Variations in the cut-off levels and the diagnostic performance of IL-6 and CRP among the previous studies and this study might be due to the variations in the number of neonates enrolled in the study, the type of patient categories (culture proven sepsis, clinical sepsis or both) used for cut-off level calculations, the age of the neonates and the number of low birth weight infants included in the study. Sampling time is also another important issue in the correct diagnosis of neonatal sepsis. This issue was clearly demonstrated in our study with marked differences in CRP concentrations, differences in optimal cut-off values and diagnostic performance in the five days serial measurements. The peak time of CRP concentration was observed in the second day of inflammatory response and afterwards the concentration showed a declining trend in the third, fourth and fifth days of measurements. In most other previous studies optimal cut-off values calculated only based on a single measurement of CRP. There might also be differences for IL-6 and CRP results between different laboratories, especially if they use different assays. Line 356-367.

Comment 2: Line 340: Specify what is meant by “The combination of IL-6 and CRP has better diagnostic performance than the use of IL-6 or CRP alone”. This combination will give a better sensitivity, but at a cost of a decreased specificity and PPV.

Response: The diagnostic performance of these combinations is higher than the use of IL-6 or CRP alone as these combinations showed markedly increased sensitivity and NPV with a slightly decreased specificity and PPV compared with the use of IL-6 or CRP alone for the diagnosis of neonatal sepsis. Line 352-355.

Comment 3: Additionally, changes may be needed after recalculating data.

Response: We have made all the necessary changes after recalculating the data as indicated in the above responses.

Ethics:

Comment 4: There is no declaration that the study was performed according to the rules of the Declaration of Helsinki and no information on approval from an ethics committee. This must be added according to Instructions of Authors.

Response: The study was conducted following the ethical considerations according to Medicinal Products law and central Ethics committee (Stellungnahme der Zentralen Ethikkommission "Die (Weiter-)Verwendung von menschlichen Körpermaterialien für Zwecke medizinischer Forschung“ (2003), and § 24 MPG II1 (2010). Only the information needed for this study was extracted and coded as is required for answering the research question. Any identifying information from the data set was removed by data controller before further usage and analysis. Safeguards were in place for appropriate and ethical use of the data. Confidentiality clauses were explicitly specified for those who do the data extraction. Permission to conduct this study was also obtained from Institute of Medical Microbiology and Epidemiology of Infectious Disease; Institute of Laboratory Medicine, Clinical Chemistry and Molecular Diagnostics; and Department of Neonatology, Faculty of Medicine, University of Leipzig. Line 97-107.

Reviewer 2 Report

Tessema and colleagues present a retrospective study with 899 neonates in which they analyzed IL-6 and CRP levels for the detection of neonatal sepsis. They investigated the optimal cut-off levels and performance of both proteins.

This is a solid manuscript on a very important and relevant topic. The high number of samples is impressive and the analysis is done thoroughly. I have only some minor concerns that the authors should address:

  • Lines 70-79: It should be made clear that IL-6 induces the expression of CRP.
  • Line 93: “Il-6” should be “IL-6”
  • Line 107: “Il-6” should be “IL-6”
  • Lines 117-122: Could the authors comment whether the device that is used for detection of IL-6 and CRP influences their results? Are they directly comparable to other laboratories which use different assays?
  • Table 2: I would present the IL-6 levels as ng/ml
  • Line 186: should be “IL-6”, not “IL- 6”
  • Lines 310: CONS is defined here, although CoNS has been used throughout the manuscript. It is unclear why this is defined here and what the difference to CoNS is (if there is any)

Author Response

Reviewer 2

Tessema and colleagues present a retrospective study with 899 neonates in which they analyzed IL-6 and CRP levels for the detection of neonatal sepsis. They investigated the optimal cut-off levels and performance of both proteins.

This is a solid manuscript on a very important and relevant topic. The high number of samples is impressive and the analysis is done thoroughly. I have only some minor concerns that the authors should address:

Response:  The reviewer’s comments are greatly appreciated. The manuscript has been revised according to the reviewer’s comments and suggestions.

Comment 1: Lines 70-79: It should be made clear that IL-6 induces the expression of CRP.

Response: This paragraph is revised and the issue raised by the reviewer is addressed by adding a sentence, IL-6 has a role in the production of CRP from the liver. Line 71-80.

Comment 2: Line 93: “Il-6” should be “IL-6”

Response: Corrected. Line 95.

Comment 3: Line 107: “Il-6” should be “IL-6”

Response: Corrected. Line 123.

Comment 4: Lines 117-122: Could the authors comment whether the device that is used for detection of IL-6 and CRP influences their results?

Response: The measurements are performed on automated analyzers, which are common in most laboratories. We work according to the German RiliBÄK for laboratory analysis and have an accreditation. “the device” should therefore not influence the results.

Are they directly comparable to other laboratories which use different assays?

Response: There might be differences for IL-6 and CRP results between different laboratories, especially if they use different assays.  Line  366-367.

Comment 5: Table 2: I would present the IL-6 levels as ng/ml

Response: The IL-6 levels are recorded in the laboratory and reported as pg/ml. So we want to present the IL-6 levels similar to the routine practice for reporting and documenting the results in the laboratory records.

Comment 6: Line 186: should be “IL-6”, not “IL- 6”

Response:  It is removed.  Line 238.

Comment 7: Lines 310: CONS is defined here, although CoNS has been used throughout the manuscript. It is unclear why this is defined here and what the difference to CoNS is (if there is any)

Response: Corrected. CoNS is defined in its first appearance in the text. Line 117 and 371.

Reviewer 3 Report

Paper by Tessema et al. aims to fix the optimal cut-off levels and performance of IL-6 and CRP for the early diagnosis of neonatal sepsis. 

This retrospective study includes A total of 899 neonates 104 with culture proven sepsis, 160 defined as clinical sepsis based only on levels of CRP >10 and negativity of culture in at least one of 5 days considered and 625 defined as controls on the bases of negativity of hemoculture and <10 CRP levels.

My major concerns regarding this study is the following:

I agree that  the sensitivity and specificity of blood culture is overall low, but CRP level is a very sensitive but not specific sign of inflammation. Modality of delivery, localized infections, not infectious disease, neonatal physiological jaundice might be causes of sustained CRP increase in the first 72 hours after birth. As reported in the paper, among clinical sepsis cases 54.4% were early onset sepsis cases, whereas the majority of late onset were "culture proven sepsis". So a larger set of information should be used to define clinical sepsis. I understand that in a retrospective study is difficult to obtain information on signs and symptoms of neonates, gestational age and mode of delivery (as stated in limitation section) but authors have the possibility to include in their study a larger set of laboratory data that can be used to refine "clinical sepsis" definition. As an example the very large number of useful data that one can obtain  from automated blood count and enzymology (e.g. LDH)  combined with PCR and analyzed by multiple regression models  might be useful in better define "clinical sepsis". 

So, definition of optimal cut-off levels of "clinical sepsis" should be revised in this light. the same methodological approach to the definition of controls is desirable.

A minor suggestion regards the definition of ethnicity of the neonates. IL-6 production is influenced by the genetic variations in the promoter (e.g. -174GC) and regulator regions of IL6 gene. The frequencies of these genetic variations are different in different Caucasian populations as well in persons of Asian or African ethnicity. The homogeneity or dis-homogeneity of the group of patients might be considered to evaluate the results obtained.

In conclusion this paper is of quality but the above suggested improvements should be necessary for pubblication

Author Response

Reviewer 3

Paper by Tessema et al. aims to fix the optimal cut-off levels and performance of IL-6 and CRP for the early diagnosis of neonatal sepsis. 

This retrospective study includes A total of 899 neonates 104 with culture proven sepsis, 160 defined as clinical sepsis based only on levels of CRP >10 and negativity of culture in at least one of 5 days considered and 625 defined as controls on the bases of negativity of hemoculture and <10 CRP levels.

Comment 1: My major concerns regarding this study is the following:

I agree that  the sensitivity and specificity of blood culture is overall low, but CRP level is a very sensitive but not specific sign of inflammation. Modality of delivery, localized infections, not infectious disease, neonatal physiological jaundice might be causes of sustained CRP increase in the first 72 hours after birth. As reported in the paper, among clinical sepsis cases 54.4% were early onset sepsis cases, whereas the majority of late onset were "culture proven sepsis". So a larger set of information should be used to define clinical sepsis. I understand that in a retrospective study is difficult to obtain information on signs and symptoms of neonates, gestational age and mode of delivery (as stated in limitation section) but authors have the possibility to include in their study a larger set of laboratory data that can be used to refine "clinical sepsis" definition. As an example the very large number of useful data that one can obtain from automated blood count and enzymology (e.g. LDH)  combined with PCR and analyzed by multiple regression models  might be useful in better define "clinical sepsis". So, definition of optimal cut-off levels of "clinical sepsis" should be revised in this light. the same methodological approach to the definition of controls is desirable.

Response: The reviewer’s comments are greatly appreciated. As mentioned in the limitation part of the article, the data from automethod blood culture, enzymology and PCR are not available in the laboratory records for these study participants to more accurately define clinical sepsis and control groups. The unavailability of these specific data is now included in the limitation part. Line 398-400.

Due to the above mentioned limitations to accurately define clinical sepsis, authors removed the optimal cut-off levels of clinical sepsis from Table 3 and other parts of the article, and instead we calculated the SN, SP. PP and NP of IL-6 and CRP based on their optimal cut-off values defined using proven sepsis and presented in Table 4. Line 256-284.

Comment 2: A minor suggestion regards the definition of ethnicity of the neonates. IL-6 production is influenced by the genetic variations in the promoter (e.g. -174GC) and regulator regions of IL6 gene. The frequencies of these genetic variations are different in different Caucasian populations as well in persons of Asian or African ethnicity. The homogeneity or dis-homogeneity of the group of patients might be considered to evaluate the results obtained.

Response:  Information about the ethnicity of the study participants is not available in the laboratory records. The unavailability of these specific information is now included in the limitation part. Line 400.

Comment 3: In conclusion this paper is of quality but the above suggested improvements should be necessary for publication

Response: We have re-checked the availability of the useful data suggested by the reviewer to better define the clinical sepsis and control groups in the laboratory records, however, these data are not documented in the laboratory records. Therefore, we tried to improve the paper by addressing those issues in the limitation part of the paper, and by excluding less accurate information from the paper like the optimum cut off-values of these markers based on the patient categories including clinical sepsis from Table 3. Line 256-284.

Round 2

Reviewer 1 Report

Suggests following corrections.

Introduction, line 58-59. The sentence has not been corrected as requested. Please correct the sentence to: Moreover, the delay in turnaround time for a blood culture result also…

Materials and Methods: Line: 142 and 145. Detection limit only refers to the lower value in the measuring range. Please change, by omitting the maximum value.

Table 2: Range should be an interval between lowest and highest value. Please also specify the lowest value.

Line 216: Change from “mean” to “median”.

Table 4: Please check your calculations. For example, SN larger for CRP 1 alone compared to CRP 1 combined with IL-6?

Results: Line 187-195. Please also add IL-6 and CRP median concentrations for the control group and the group with clinical sepsis. Median concentrations for the proven sepsis group are presented in table 2 (“total”), but there is no data from the other two groups.

Discussion and Abstract:

The term “diagnostic performance” should not be used when comparing combined parameters (IL-6 and CRP) with CRP or IL-6 alone. To be able to make conclusions about with strategy of testing that has the best diagnostic performance you should perform a ROC curve analysis using two parameters (usually by logistic regression) and compare the AUC with the AUCs found using only one parameter. So, I suggest that you either perform this analysis or omit the sentence Line 33: “The combination of IL-6 and CRP has better diagnostic performance than the use of IL-6 or CRP alone.” and Line 409: “The combination of Il-6 and CRP….”. Please also remove “marked” and “slight” from the sentence on line 32 in the abstract. You also have to reformulate your conclusion in the abstract and discussion, saying that the recommendation is based on increased sensitivity, that is, to minimize the risk of missing any cases of sepsis.

Line 391-394. This is a valid and important conclusion, which preferably can be added to the conclusion section and abstract.

Author Response

Suggests following corrections.

Comment 1:  Introduction, line 58-59. The sentence has not been corrected as requested. Please correct the sentence to: Moreover, the delay in turnaround time for a blood culture result also…

Response:  Corrected: Line 63.

Comment 2: Materials and Methods: Line: 142 and 145. Detection limit only refers to the lower value in the measuring range. Please change, by omitting the maximum value.

Response: Corrected: Line 144 and 147.

Comment 3: Table 2: Range should be an interval between lowest and highest value. Please also specify the lowest value.

Response: Corrected: Line 204 -206.

Comment 4: Line 216: Change from “mean” to “median”.

Response: Corrected: Line 211.

Comment 5: Table 4: Please check your calculations. For example, SN larger for CRP 1 alone compared to CRP 1 combined with IL-6?

Response: We have checked the calculations and it is correct. As the SN for IL-6 is very low, SN is larger for CRP 1 alone compared to CRP 1 combined with IL-6.

Comment 6:  Results: Line 187-195. Please also add IL-6 and CRP median concentrations for the control group and the group with clinical sepsis. Median concentrations for the proven sepsis group are presented in table 2 (“total”), but there is no data from the other two groups.

Response: Corrected: Line 187 - 194.

Discussion and Abstract:

Comment 7: The term “diagnostic performance” should not be used when comparing combined parameters (IL-6 and CRP) with CRP or IL-6 alone. To be able to make conclusions about with strategy of testing that has the best diagnostic performance you should perform a ROC curve analysis using two parameters (usually by logistic regression) and compare the AUC with the AUCs found using only one parameter. So, I suggest that you either perform this analysis or omit the sentence Line 33: “The combination of IL-6 and CRP has better diagnostic performance than the use of IL-6 or CRP alone.” and Line 409: “The combination of Il-6 and CRP….”.

Response: The sentence is modified removing the term “diagnostic performance” as follow the combination of IL-6 and CRP demonstrated increased sensitivity. Line 32-33 and 387-388.

Comment 8: Please also remove “marked” and “slight” from the sentence on line 32 in the abstract.

Response: “marked” and “slight” are removed from the sentence. Line  31.

Comment 9: You also have to reformulate your conclusion in the abstract and discussion, saying that the recommendation is based on increased sensitivity, that is, to minimize the risk of missing any cases of sepsis.

Response: The conclusion is reformulated in the abstract and discussion as commented. Line 37-38 and 391-392.

Comment 10: Line 391-394. This is a valid and important conclusion, which preferably can be added to the conclusion section and abstract.

Response: It is added to the conclusion section and abstract. Line 34-35, 38-39, 388-389, and 392-393.

Reviewer 3 Report

Authors have  included in the "limitation Section" the critical points indicated in my review report and have reshaped table 3 considering that evidences obtained have been weakened by the lack of additive clinical and laboratory data.

On the other hand improvements introduced in the revised version of the paper allow to recommend paper for publication.

Author Response

Comments and Suggestions for Authors

Authors have  included in the "limitation Section" the critical points indicated in my review report and have reshaped table 3 considering that evidences obtained have been weakened by the lack of additive clinical and laboratory data.

On the other hand improvements introduced in the revised version of the paper allow to recommend paper for publication.

Response:  We are happy that the revised version of the article satisfied the reviewer.